# Impact of Forest Conversion to Agriculture on Hydrologic Regime in the Large Basin in Vietnam

**Nguyen Cung Que Truong [1,\*], Dao Nguyen Khoi [2], Hong Quan Nguyen [3,4] and Akihiko Kondoh [1]**

[1] Center for Environmental Remote Sensing, Chiba University, Chiba 263-8522, Japan; kondoh@faculty.chiba-u.jp

[2] Faculty of Environment, University of Science, Vietnam National University—Ho Chi Minh City, Ho Chi Minh 700000, Vietnam; dnkhoi@hcmus.edu.vn

[3] Institute for Circular Economy Development, Vietnam National University—Ho Chi Minh City, Ho Chi Minh 700000, Vietnam; nh.quan@iced.org.vn

[4] Center of Water Management and Climate Change, Institute for Environment and Resources, Vietnam National University—Ho Chi Minh City, Ho Chi Minh 700000, Vietnam

\* Correspondence: cungque@gmail.com; Tel.: +81-43-251-1111

**Abstract:** Deforestation due to agricultural land expansion occurred greatly during 1994 to 2005 with a high proportion of forests being converted into agriculture in the upstream Dong Nai river basin in Vietnam. Most of these conversions included expansions of coffee plantations in Dak Lak and Lam Dong provinces, which are in the world's Robusta coffee production area. The aim of this study is to quantify the impact on the water cycle due to the conversion of forest to coffee plantations in a tropical humid climate region by the application of a hydrological model: soil and water assessment tool (SWAT). The model was calibrated with climate data from 1980–1994, validated with climate data from 1995–2010, and verified with statistical indicators such as Nash–Sutcliffe efficiency (NSE), percent bias (PBIAS), and ratio of the root mean square error (RSR). The simulations indicated that forest conversions into agriculture (expansion of coffee plantations) had significantly increased surface runoff (SUR) while actual evapotranspiration (ET), soil water content (SW), and groundwater discharge (GW) decreased. These changes are mainly related to the decrease in infiltration and leaf area index (LAI) post land cover changes. However, the soil was not thoroughly destroyed after deforestation due to the replacement of the lost forest with crops and vegetation. Therefore, changes in infiltration were marginal and not sufficient to bring large changes in the annual flow. Higher reductions in ET and SW were proposed, resulting in reduced streamflow in the dry season at the basin where the proportion of agricultural land was higher than the forest cover. Besides the plantation expansion, which resulted in streamflow reductions in the dry season, an existing problem was over-irrigation of coffee plantations that could likely deplete groundwater resources. Hence, balancing economic benefits by coffee production and mitigating groundwater depletion issues should be prioritized for land use management in the study area.

**Keywords:** Dong Nai river basin; LUCC; flow regime; SWAT; coffee plantation

## 1. Introduction

The population of Vietnam has increased rapidly since 1960 and reached nearly 96 million by 2017. The Vietnamese government initiated a series of economic reforms in the mid-1980s, which were aimed at stimulating economic growth, the most remarkable policy being the New Economic Zones program. This policy resulted in large-scale displacement of residents to uninhabited areas due to expansion of the agricultural land. These factors led to the conversion of land use and land cover in Vietnam after mid-1980s [1]. Furthermore, since 1998, the Vietnamese government implemented multiple national No programs for poverty reduction and programs for socio-economic development in

mountainous areas; for example, Decision No.133/1998/QD-TTg dated 23 July 1998, Decision No.135/1998/QD-TTg dated 31 July 1998, and Decision No.143/2001/QD-TTg dated 27 September 2001. These decisions highlighted migration and land reclamation as important projects of the program, which in turn led to a major disturbance in land use.

As described in [2], forested area declined sharply between 1994 and 2005 throughout the study area, which was the upstream Dong Nai river basin (UDNB) (Figure 1), due to the conversion to agricultural land caused by exponential population growth. The proportions of forest and agricultural land in 1994 were 73% and 23%, respectively, which subsequently changed to 51% and 40%, respectively, in 2005. It led to an immediate increase in total discharge, streamflow, and abundant, low, and scanty runoffs. This study was initiated to complement the results of Truong et al., (2018) [2], and analyze the elements, which alter the streamflow, through the evaluation of the changes in the hydrological components by using the soil and water assessment tool (SWAT) model [3].

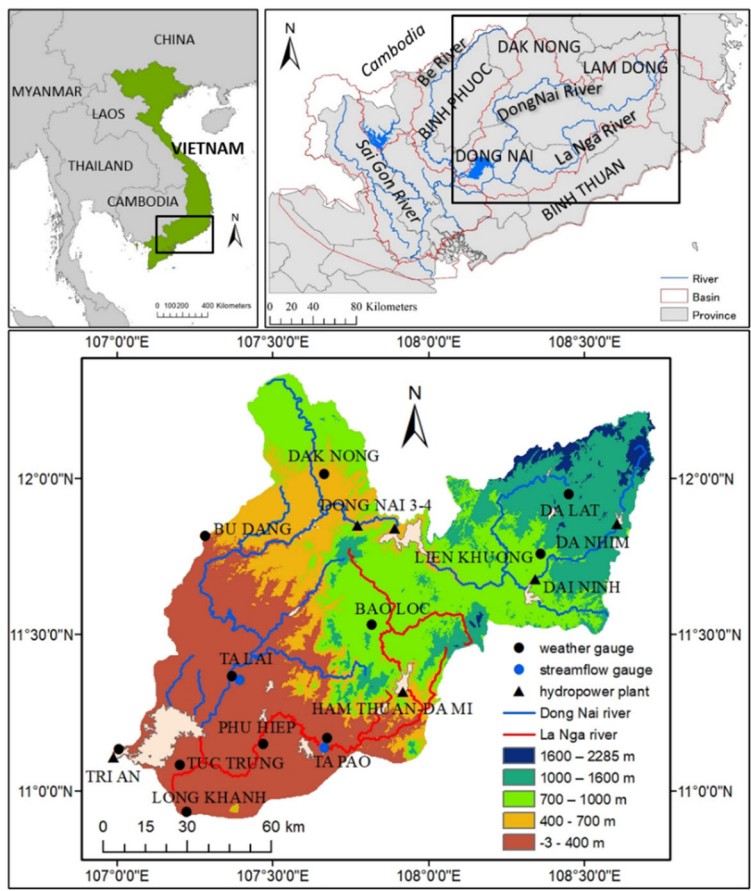

**Figure 1.** The UDNB with weather, stream gauges and hydropower-plant locations. (Modified from [2]).

Research on the impact of land use/land cover change (LUCC) on hydrology has increased since the 1960s. Several reviews [4–9] have focused on the changes in the annual streamflow and neglected the changes in flow regime. These studies revealed that increased forest cover reduced the annual streamflow and vice versa; additionally, the total annual water yield increased with the increase in lost forest percentage [10]. However, the studies on the effects of LUCC in large tropical river basins could not estimate similar relationships [6], whereas water resource management usually requires information on regional (> 1000 km²) or large-scale catchments [11]. Furthermore, the effect of forest cover change on flooding, particularly in developing countries, is an ongoing discussion [12,13]. Bradshaw et al., (2007) [12] shows that flood frequency is positively correlated with natural forest area loss by using generalized linear modeling. Yet, Van Dijk et al., (2009) [13]

reanalyzed the data used in [12] and suggested that the removal of trees does not affect large flood events. It is also known that forests present finite capabilities to retain large amounts of precipitation, especially during extreme rainfall events, even if the forest cover percentage is significantly high [14,15]. Defining a threshold above which forest cover is no longer effective in reducing a flood is a challenge in forest hydrology [16].

The following summaries of the study in a large tropical rainforest river basin present different perspectives on the overall impact of vegetation on streamflow. Few catchment studies indicated that the annual discharge increased with forest-to-crop expansion due to low evapotranspiration and infiltration rates.

Thanapakpawin et al., (2006) [17] assessed hydrological regimes with land use change at the Mae Chaem river basin (3853 km2, elevation 255–2565 m) in northwestern Thailand; this was achieved by three forest-to-crop expansion scenarios and a crop-to-forest reversal scenario. The results showed that unregulated runoff increased with the conversion from forest to crops, owing to decreasing evapotranspiration and irrigation diversion directly influencing discharge magnitude and significantly varying water yields. Similarly, Costa et al., (2003) [18] evaluated the effect of land use changes on discharge in the upper Tocantins basin (175,360 km²). The authors estimated the annual and seasonal mean discharges with two datasets: (1) agricultural land use of 30.2% (1960) and climate data during 1949–1968; and (2) agricultural land use of 49.2% (1995) and climate data during 1979–1998. Although precipitation did not significantly differ between the two periods, the annual discharge increased by 24%, rainy season discharge increased by 28%, and seasonal peaks occurred about one month earlier. Post land cover changes, reduced infiltration, increased the rainy season surface flow and actual evapotranspiration reduction increased the discharge throughout the year.

Studies conducted in Vietnam also concluded that forest gain can decrease annual runoff while forest loss affects in the opposite way. Nguyen et al., (2014) [19] simulated water discharge in the Srepok watershed in the central highlands by using the SWAT hydrological model. The increase in forest cover area from 50.45% in 2000 to 79.59% in 2010 resulted in the reduction in surface runoff percentage by half. The impact of land use changes on hydrological processes in the Be River catchment was investigated with SWAT by Khoi and Suetsugi (2014) [20]. The results indicated that deforestation increased the surface runoff and soil water content (over 10%) while actual evapotranspiration, water yield, and annual flow increased marginally (approx. 1%).

In contrast, other studies showed that vegetation has no impact on streamflow. Wilk et al., (2001) [21] used the Hydrologiska Byråns Vattenbalansavdelnin (HBV) hydrology model, which is a conceptual model that simulates daily discharge with input data of daily rainfall and temperature, and monthly estimates of potential evaporation to determine the rainfall or change in runoff regime after a decrease in forest cover from 80% (1965) to 27% (1992) at the Upper Nam Pong Basin (area 12,100 km², elevation 300–1400 m) in northeastern Thailand. However, no detectable trends in river discharge were observed possibly due to a significant number of remaining trees and secondary growth on agricultural land. Beck et al., (2013) [22] examined the effects of afforestation on the streamflow for 12 mesoscale catchments (area 23–346 km²) in Puerto Rico. However, the correlation between changes in the forest area and changes in streamflow was insignificant. The three possible reasons were data errors, heterogeneity in catchment response, and streamflow generation in the headwater areas, whereas changes in forest area mainly occurred in the drier lowlands. Similarly, the spatial variations of LUCC on streamflow were estimated in the study by Liu et al., (2020) [23].

In addition, the effects of logging methods on water yield and streamflow in the tropical forest watershed were conducted by Malmer (1992) [24]. Malmer observed that a combination of burning and no soil disturbance substantially increased the water yield as compared to cases that experienced soil disturbance and loss of infiltrability. Additionally, it led to high-speed runoff during storms in Mendolong, Malaysia, which included six catchments with areas varying from 3.4 to 18.2 ha.

The purpose of this study is to quantify the impacts on the water cycle due to the conversion of forest to coffee plantations in the UDNB, a large tropical rainforest basin, by assessing the changes in the water balance components, such as the actual evapotranspiration (ET—actual evapotranspiration during the time step; measured in mm), surface runoff (SUR—surface runoff contribution to streamflow during the time step; measured in mm $H_2O$), groundwater discharge (GW—groundwater contribution to streamflow during the time step; measured in mm), and soil water content (SW—amount of water in soil profile at the end of the time period; measured in mm). Effective water resource management and strategic planning needs to consider the effect of LUCC and water availability, particularly in the agricultural sector. The findings from this study will be significant to decision-makers working for integrated river basin management for the development of land use adaptation and mitigation strategies.

## 2. Materials and Methods

### 2.1. Study Area

UDNB is located in Vietnam in the central highlands, which is a tropical humid zone receiving southwest monsoons; 90% of the annual rainfall (the average rainfall from 1993 to 2012 was 2415 mm/year) occurs during the rainy season (May–October) while the dry season (November–April) receives the remaining rainfall. Since the basin extends from the mountains to the lower plains, the temperature varies significantly throughout. The average temperature ranges between 18 and 26 °C [2] (Figure 2).

The main stem of the Dong Nai River originates from the high hill (elevation 1000 m to 2000 m) north of Lam Dong province, where it is called Da Nhim River. Its flow course initially follows the southwestward direction and later turns to the west forming the border line between Lam Dong and Dak Nong provinces. Thereafter, Dong Nai River heads to the southeast and crosses Dong Nai province in the southwest. The La Nga River originating from Lam Dong province lies to the south of the basin and merges into the Dong Nai River before it flows into the Tri An reservoir [25] (Figures 1 and 3). The high-gradient streams indicated a steep slope and rapid flow of water.

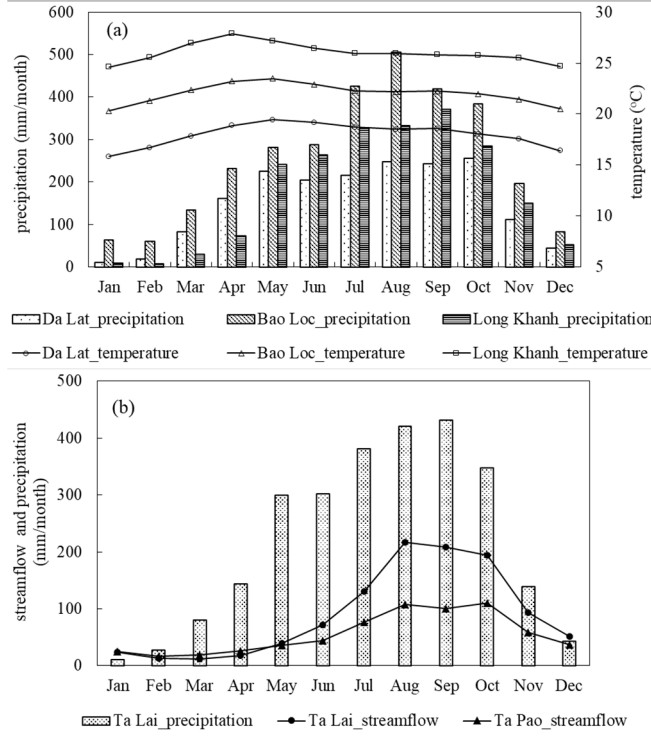

**Figure 2.** Mean monthly temperature and precipitation at different elevations in three stations Da Lat (>1000 m), Bao Loc (400~1000 m), and Long Khanh (<400 m) (**a**). Mean monthly streamflow

compared to precipitation at Ta Lai and Ta Pao stations (**b**). (Precipitation data are not available at Ta Pao station).

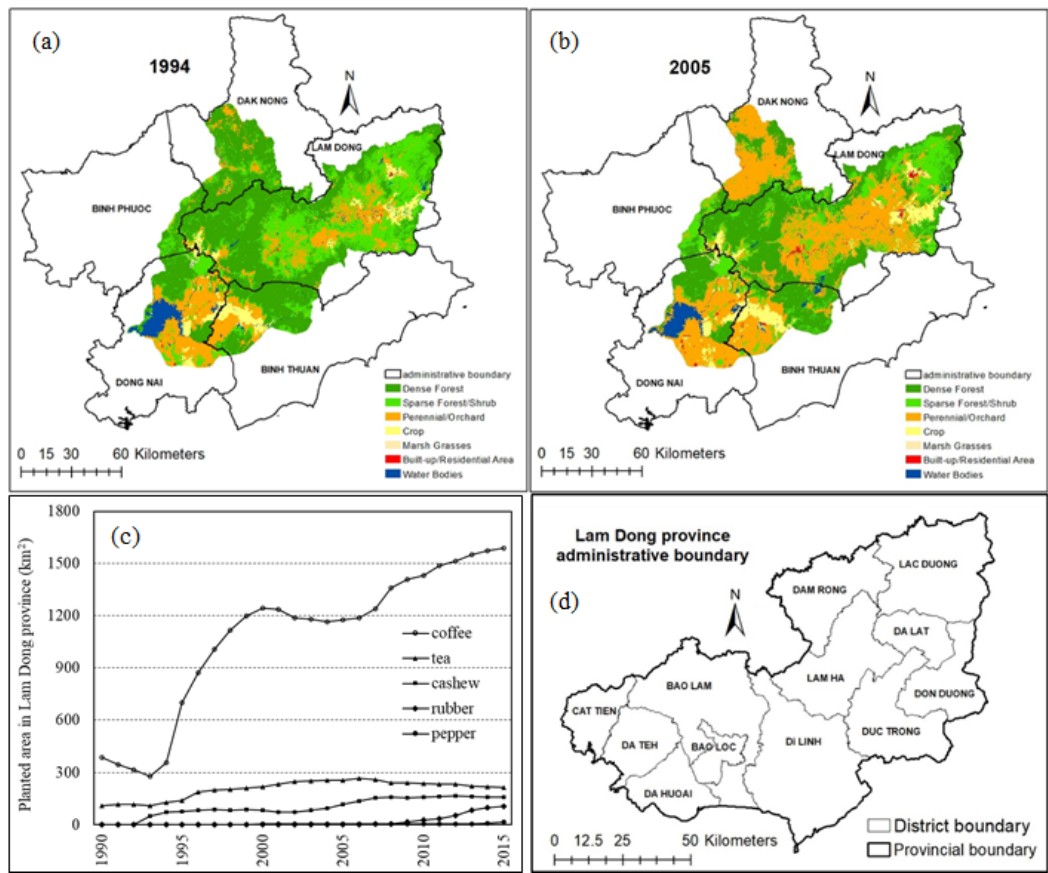

**Figure 3.** Land-use changes of UDNB in 1994 and 2005 (**a**,**b**); Trends in the planted areas of multi-year industrial crops (**c**) and administrative boundaries of Lam Dong province (**d**) (Data source: Lam Dong Statistical Office).

### 2.2. Land Use/Land Cover Change in the Study Area

According to [2], land use changed intensely between 1994 to 2005; a high proportion (area approximately 3343 km², which accounts for 31.12% forest cover) of forest land was converted to agricultural land. Dense forest and sparse forest/shrub of the forest area constituted 73% whereas agricultural land, which included perennial orchard and crops constituted 24% of the total area (14,706 km²) in 1994. The forest cover decreased to 51% in 2005 while agriculture land increased to 45%. The majority of the land use changes were in Lam Dong and Dak Nong provinces. Figure 2 shows land cover maps of the UDNB for 1994 and 2005 classified from Landsat 5 TM images (path 124 row 52, 7 January 1994; and 22 February 2005) (reference from [2]).

The coffee sector in Vietnam grew exponentially throughout the 1990s due to government mandates and incentives in the form of favorable credit, subsidized inputs, and low-cost land for exporting crops. Since the early 2000s, Vietnam became the world's second-largest coffee producer with 30% contribution to the GDP of the central highlands, the largest Robusta coffee production area worldwide [26]. In this region, Lam Dong and Dak Lak (Dak Lak was divided into two separate provinces: Dak Lak and Dak Nong since 2004) are the largest coffee producing provinces, which cover nearly 3442 km² (68%) of the total 5065 km² area of coffee plantations of Vietnam in 2003 [27]. The coffee plantations in Lam Dong province increased by 819.43 km² (from 355.95 km² to 1175.38 km²) (Figure 2). Five districts including Bao Lam, Bao Loc, Duc Trong, Di Linh, and Lam Ha observed major land use changes in Lam Dong province and comprised the largest coffee

plantations (over 90% of Lam Dong's total coffee plantations) (Table 1). However, the area under coffee plantations increased from approximately 1500 km2 in 1994 to 2400 km2 in 1999 and subsequently to 2800 km² in 2001 (accounting for 51.25% of agricultural land and 14.28% of the total area) [28]. Thus, the majority of land use conversions in the study area accounted for expansion of coffee plantation.

**Table 1.** Coffee Planted Areas by Districts in Lam Dong Province (km²).

| Year | 1995 | 1999 | 2000 | 2001 | 2002 | 2003 | 2004 | 2005 |
|------|------|------|------|------|------|------|------|------|
| Lac Duong | 2.01 | 14.02 | 16.55 | 16.55 | 16.70 | 16.52 | 13.77 | 10.40 |
| Dam Rong | - | - | - | - | - | - | - | 30.94 |
| Da Lat | 8.03 | 34.29 | 38.41 | 36.76 | 34.4 | 34.6 | 33.12 | 33.45 |
| Don Duong | 4.48 | 14.60 | 17.02 | 13.56 | 10.11 | 8.56 | 8.56 | 7.96 |
| Lam Ha | 217.50 | 333.53 | 344.36 | 344.94 | 346.30 | 343.22 | 340.17 | 320.61 |
| Duc Trong | 18.07 | 92.26 | 101.61 | 109.54 | 93.79 | 86.03 | 80.91 | 78.79 |
| Di Linh | 229.63 | 373.32 | 382.92 | 378.34 | 362.96 | 361.92 | 361.62 | 361.63 |
| Bao Lam | 183.63 | 262.36 | 263.64 | 262.94 | 257.94 | 257.08 | 257.69 | 259.47 |
| Bao Loc | 30.55 | 61.44 | 64.41 | 72.74 | 68.86 | 69.82 | 68.27 | 69.39 |
| Cat Tien | 2.00 | 2.09 | 1.70 | 0.57 | 0.55 | 0.30 | 0.30 | 0.00 |
| Da Huoai | 3.31 | 10.11 | 9.34 | 6.3 | 3.58 | 1.26 | 0.87 | 0.62 |
| Da Teh | 1.20 | 2.13 | 3.63 | 3.48 | 2.52 | 2.37 | 2.12 | 2.12 |
| Total | 701.04 | 1200.15 | 1243.59 | 1237.39 | 1190.01 | 1181.68 | 1167.4 | 1175.38 |

Data source: Lam Dong Statistical Office.

### 2.3. The Soil and Water Assessment Tool (SWAT) Model

Hydrological modeling was selected to quantify the impacts of LUCC and climate variability on the hydrological parameters. Globally, many researchers have confirmed the high performance and algorithms of SWAT [1,29–36]. SWAT model [36,37] is a continuous, long term, distributed parameter model that was developed to predict the long-term impacts of land management practices on water, sediment and agricultural yields in large complex watersheds with varying soils, and land use and management conditions. SWAT requires specific information about weather, soil properties, topography, vegetation, and land management to model the physical processes associated with water movement, sediment movement, etc.

A watershed is divided into multiple subwatersheds, which are further subdivided into hydrological response units (HRUs) that comprise homogeneous land use, land management, and topographical and soil characteristics [3]. The hydrologic cycle simulated by SWAT is based on the water balance equation:

$$SW_t = SW_0 + \sum_{i=1}^{t} \left( R_{day} - Q_{surf} - E_a - w_{seep} - Q_{gw} \right) \tag{1}$$

where $SW_t$ is the final soil water content (mm H₂O), $SW_0$ is the initial soil water content on day $i$, $t$ is the time, $R_{day}$ is the amount of precipitation on day $i$, $Q_{surf}$ is the amount of surface runoff on day $i$, $E_a$ is the amount of evapotranspiration on day $i$, $w_{seep}$ is the amount of water entering the vadose zone from the soil profile on day $i$, and $Q_{gw}$ is the amount water return flow on day $i$ (mm H₂O). A detailed description of the model is given in the SWAT theoretical documentation [38].

Input Data

SWAT model requires a digital elevation model (DEM), land use/land cover, soil properties, meteorological, and observed streamflow data.

- DEM. The DEM of the basin was derived from SRTM30 data that have been published by the United States National Aeronautics and Space Administration (NASA) (Figure 1).

- Soil properties. We used the soil map of the world developed by the Food and Agriculture Organization (FAO) of the United Nations.
- Land use and land cover data. The land use maps of 1994 and 2005 with seven land cover classes, dense forest, sparse forest/Shrub, perennial/orchard, crop land, built-up/residential, marsh/grasses, and water body, which were classified from the Landsat imagery, were used (Figure 2) [2]. To assess the performance of SWAT under LUCC, the daily hydrographs for the two land use maps 1994 and 2005 with the same climatic conditions and model parameters were simulated.
- Meteorological data. Data for daily precipitation (mm), maximum and minimum temperatures (°C), solar radiation (Wm$^{-2}$), wind speed (ms$^{-1}$), and relative humidity (%) were provided by the provincial department of natural resources and environment (DONRE). The study area was located in a tropical humid zone receiving southwest monsoon; 80% of the annual rainfall occurred during the rainy season (May–October) while the dry season (November–April) received the remaining rainfall. As UDNB extends from the high hills to the low plain area, temperature and precipitation vary significantly (Figure 3).
- Streamflow data. Daily streamflow data (m$^3$/s) at Ta Lai station from 1 January 1987 to 31 December 2010 were collected and data at Ta Pao station from 1 January 1980 to 31 December 2010 (Figure 1) were compared with the modeled surface flow.

### 2.4. Model Setup and Performance Evaluation

In this study, we used the geographic information system interface ArcSWAT to parameterize the model. Basin delineation was implemented by delineating the stream network from SRTM30 gridded DEM data, and assigning Tri An dam as the outlet of the study basin. The basin was divided into 111 sub-basins (SB) (Figure 3). Later, 979 HRUs were created based on the map combinations of land use, soil, and slope map. The observed meteorological data and streamflow data were used for calibration (1980/1987–1994) and performance validation (1995–2010) in the daily time step of the flow simulation. The uncertainty analysis was conducted by sequential uncertainty fitting (SUFI-2) method, which was implemented in the SWAT-CUP [39]. The final calibrated parameters for the basins of Dong Nai and La Nga rivers are presented in Table 2.

**Table 2.** Highly Sensitive Final Calibrated Parameters.

| Parameter | Description of Parameter | Range | Best Simulation | |
|---|---|---|---|---|
| | | | Dong Nai | La Nga |
| CN2.mgt | Initial SCS CN II value | −0.5–0.5 | −0.312 | −0.098 |
| CH_K2.rte | Channel effective hydraulic conductivity | −0.01–500 | 219.299 | 35.765 |
| ALPHA_BF.gw | Baseflow alpha factor | 0–1 | 1.203 | 1.003 |
| SOL_AWC.sol | Available water capacity | −0.5–0.5 | 0.942 | 0.559 |
| GWQMN.gw | Threshold water depth in the shallow aquifer for flow | 0–5000 | 1654 | 2021 |
| REVAPMN.gw | Threshold water depth in the shallow aquifer for "revap" | 0–1000 | 925 | 925 |
| GW_DELAY.gw | Groundwater delay | 0–500 | 8.219 | 33.853 |
| SOL_Z.sol | Soil depth | −0.2–0.2 | 0.000 | 0.000 |
| GW_REVAP.gw | Groundwater 'revap' coefficient | 0.02–0.2 | 0.288 | 0.072 |
| SOL_K.sol | Saturated hydraulic conductivity | −0.5–0.5 | 0.220 | 0.220 |

Ten land surface response parameters significantly affected the streamflow simulation, and therefore, they were the most sensitive parameters of the model. Among these, CN2 and SOL_AWC directly govern surface response by controlling the SUR that directly contributes to the streamflow. Curve number II (CN2), which is a function of watershed properties that includes soil type, land use and treatment, ground surface condition, and antecedent moisture conditions [40], adjusts the soil humidity for different land uses to

estimate the surface runoff. Low values of CN2 reflect decreased SUR and increased baseflow. Yet, the climate in the study area has distinct wet and dry seasons. The curve number method is able to account for this by using the empirical rainfall-runoff relationships for dry, average, and wet antecedent wetness conditions (CNI, CNII, and CNIII). Dile et al. [41] indicated that the curve number method works better in the wet season (high rainfall conditions) that it is able to be useful for hydrological simulation in tropical regions. The average streamflow in the dry season (low rainfall conditions) is small, resulting in only small errors in estimating streamflow. SOL_AWC is the volume of water available for plant uptake when the soil is at field capacity and can be estimated by determining the quantity of water released between the field capacity of soil and the point of permanent wilting [40]. Low values of SOL_AWC indicate a low soil capability to maintain its humidity, which subsequently increases the amount of water available for surface runoff and percolation. One of the parameters governing subsurface response is GW_REVAP, which controls the amount of water that will move from the shallow aquifer to the root zone as a result of soil moisture depletion, and the amount of direct groundwater uptake from deep-rooted trees and shrubs [42]. A low GW_REVAP value reflects restricted movement of water from the superficial aquifer to the root zone, while a high value indicates that the transfer rate is close to the rate of evapotranspiration.

According to Moriasi et al. [43] three quantitative statistics, Nash–Sutcliffe efficiency (NSE), percent bias (PBIAS), and the root mean square error (RSME)—standard deviation (STDEV) of measured data ratio (RSR) were used for model evaluation. Model performance can be judged based on the general performance ratings (Table 3) obtained by these values. RMSE values less than half the STDEV of measured data indicate an acceptable error range. According to that, less than 0.5 RSR value as the most stringent "very good" rating, and two less stringent ratings of 10% and 20% greater than this value for the "good" and "satisfactory" ratings, respectively [14,44]. The formulae for these statistics are given below.

$$\text{NSE} = 1 - \left[ \frac{\sum_{i=1}^{n}\left(Y_i^{obs} - Y_i^{sim}\right)^2}{\sum_{i=1}^{n}(Y_i^{obs} - Y^{mean})^2} \right] \qquad (2)$$

$$\text{PBIAS} = \left[ \frac{\sum_{i=1}^{n}\left(Y_i^{sobs} - Y_i^{sim}\right) * 100}{\sum_{i=1}^{n} Y_i^{obs}} \right] \qquad (3)$$

$$\text{RSR} = \frac{RSME}{STDEV_{obs}} = \frac{\sqrt{\sum_{i=1}^{n}\left(Y_i^{obs} - Y_i^{sim}\right)^2}}{\sqrt{\sum_{i=1}^{n}(Y_i^{obs} - Y^{mean})^2}} \qquad (4)$$

where $Y_i^{obs}$ corresponds to the $i$th observation for the constituent being evaluated, $Y_i^{sim}$ is the $i$th simulated value for the constituent being evaluated, $Y^{mean}$ is the mean of observed data for the constituent being evaluated, and n is the total number of observations.

**Table 3.** General Performance Ratings of Statistical Indices for Monthly Streamflow Simulation.

| Rating | NSE | PBIAS (%) | RSR |
|---|---|---|---|
| Very good | 0.75 < NSE ≤ 1.00 | PBIAS < ±10.00 | 0.00 ≤ RSR ≤ 0.50 |
| Good | 0.65 < NSE ≤ 0.75 | 10.00 ≤ PBIAS < ±15.00 | 0.50 < RSR ≤ 0.60 |
| Satisfactory | 0.50 < NS. E ≤ 0.65 | 15.00 ≤ PBIAS < ±25.00 | 0.60 < RSR ≤ 0.70 |
| Unstatisfactory | NSE ≤ 0.50 | PBIAS ≥ ±25.00 | RSR > 0.70 |

SWAT model for the Dong Nai upstream river basin was calibrated by comparing the simulated and observed streamflow data at two gauge stations, Ta Lai (main stream of the Dong Nai river) and Ta Pao (main stream of the LaNga river) (Figure 1). The comparison of calibration and validation data (red line) of the model with the observed data (blue line) at the Ta Lai and Ta Pao stations is shown in Figure 4. It indicates that the model

closely replicated the observational data during the calibration period. The statistical evaluations are shown in Table 4 also suggest a good agreement between the measured and simulated streamflow. PBIAS was approximately 12.92% and 0.35% for the calibration period and −7.47% and 1.20% for the validation period at the Ta Lai and Ta Pao stations, respectively. The correlation coefficients NSE were 0.88 and 0.85 at Ta Lai station, and 0.67 and 0.51 at Ta Pao station for the monthly streamflow in the calibration and the validation periods, respectively. Although the RSR for the validation period at the Ta Pao station was 0.70 (within the range of "satisfactory" benchmarks), which was comparatively less accurate, the results of the performance were still considered satisfactory, which indicates that the fundamental rainfall-runoff relationship is well documented. Thus, we affirm that these results indicate "good performance".

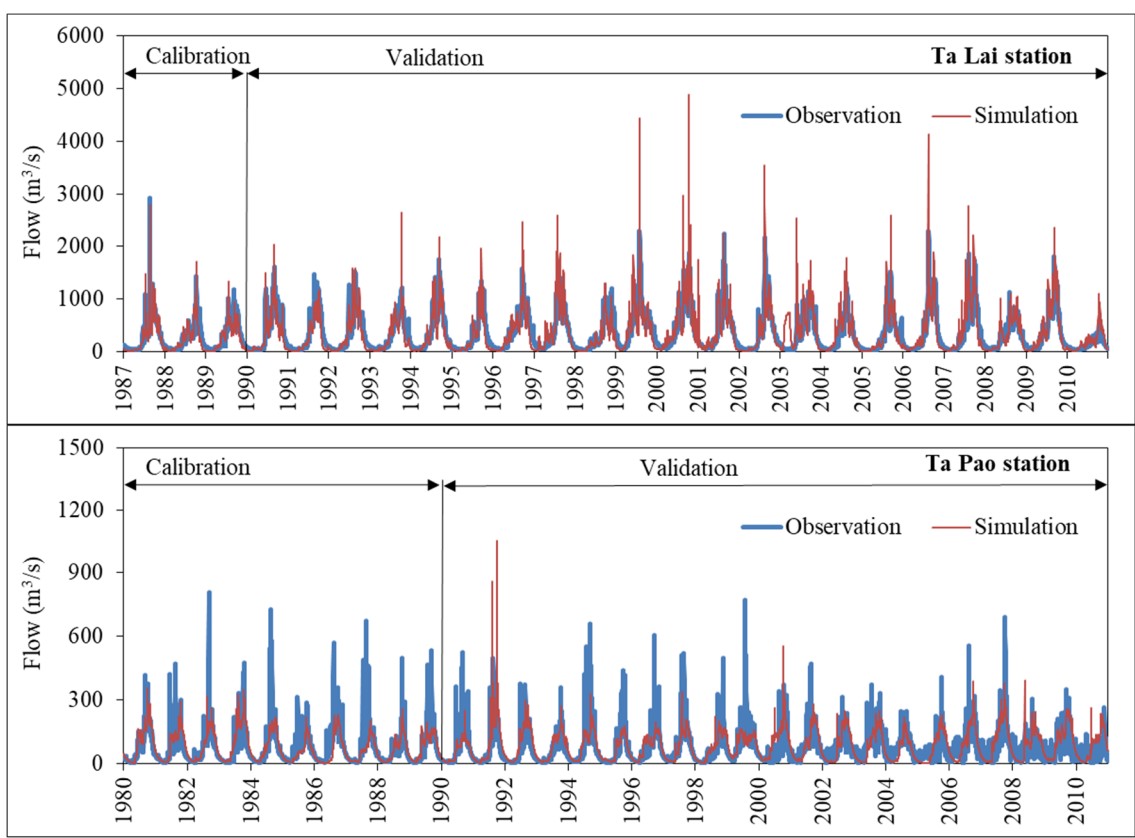

**Figure 4.** Comparison of simulation results and observed data at Ta Lai and Ta Pao stations.

**Table 4.** Model performance for calibration and validation.

| Station | Calibration (1980/1987–1994) | | | Validation (1995–2010) | | |
|---------|------|-------|------|------|-------|------|
|  | NSE | PBIAS | RSR | NSE | PBIAS | RSR |
| Ta Lai | 0.88 | 12.92 | 0.34 | 0.85 | −7.47 | 0.38 |
| Ta Pao | 0.67 | 0.35 | 0.58 | 0.51 | 1.20 | 0.70 |

## 3. Results

The results of model simulation using land use data 1994 (Landuse 1994) and 2005 (Landuse 2005) of ET, SUR, SW, and GWQ are given in Figure 5. Under the impact of LUCC, deforestation and expanding coffee plantations increased the SUR drastically by 35% at both Ta Lai and Ta Pao stations. Accordingly, the streamflow in the rainy season in both Ta Lai and Ta Pao stations increased by 5% and 81% (6% and 1% of the annual streamflow), respectively. Contrastingly, the streamflow in the dry season at Ta Pao station reduced 23% (11.06 m3/s). (Figure 6).

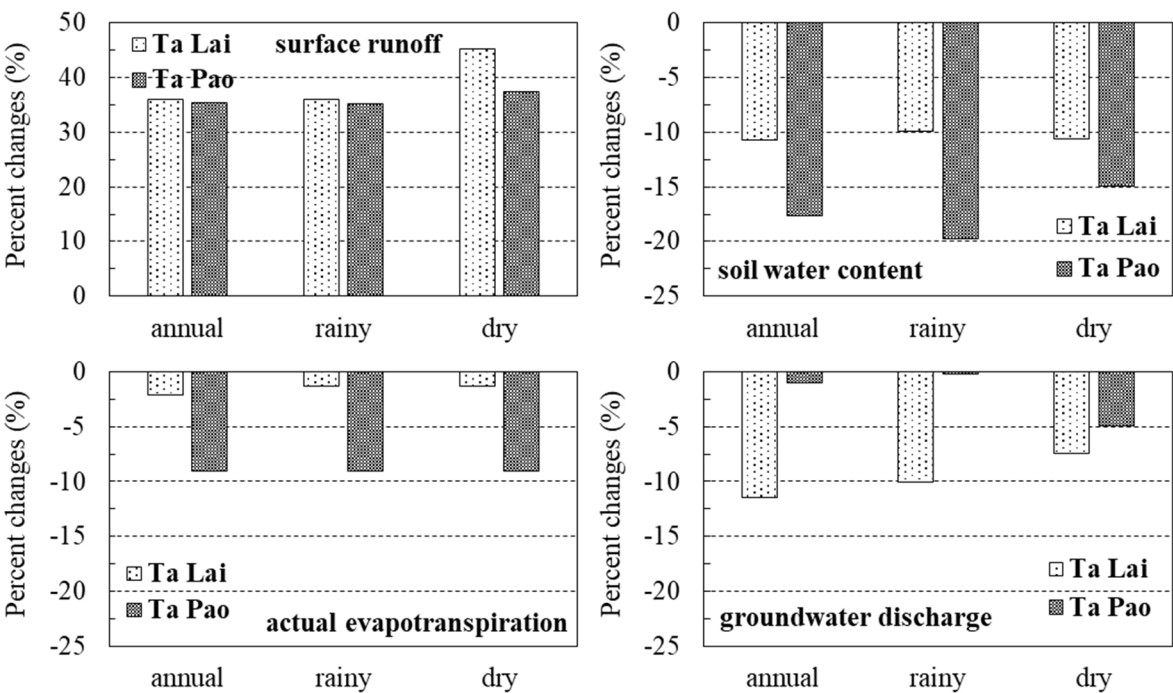

**Figure 5.** Annual and seasonal changes of hydrological components under LUCC (rainy season: May to October; dry season: November–April).

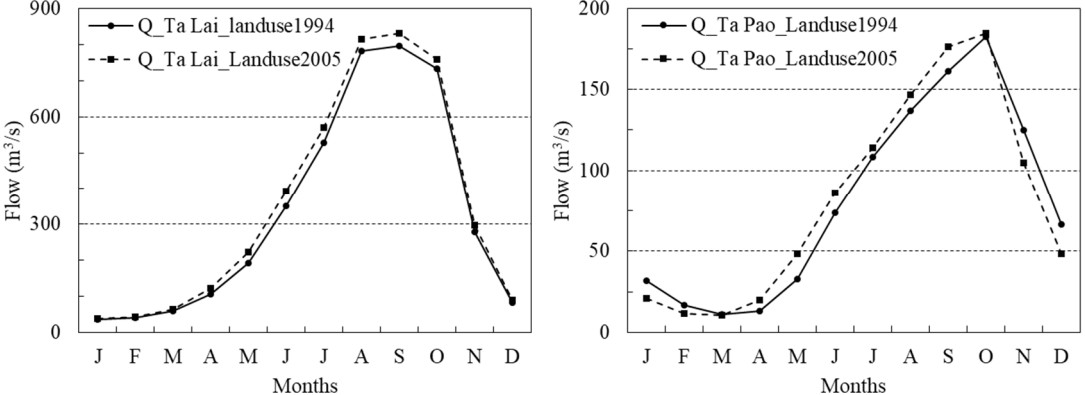

**Figure 6.** Comparison of monthly streamflow (Q–m³/s) between Landuse 1994 and 2005 at Ta Lai and Ta Pao stations.

In contrast, the ET, SW, and GWQ showed a downward trend. Annually, ET insignificantly decreased with a reduction of 2%, and SW decreased by 10% at Ta Lai station. These reductions were greater by 9% of ET and 18% of SW at Ta Pao station. GWQ was reduced significantly by 11% at Ta Lai and insignificantly by 1% at Ta Pao. ET did not differ between the dry season and rainy season, whereas SW and GWQ exhibited lower reductions in the dry season than the rainy season, and SUR in the dry season increased considerably in the rainy season.

Total ET in the SWAT model comprises the evaporation from the canopy surface, transpiration, evaporation from the soil, and groundwater evapotranspiration (Revap) [37]. Revap is the movement of water from the underlying shallow aquifer to the unsaturated zone in response to water demand for evapotranspiration. Revap can be calculated separately in regions where the saturated zone is within the root zone [45]. In this study, ET results did not include revap. Temporarily, ET was mainly controlled by soil water availability in the hot-dry season (March and April), leaf area index (LAI) in the early rainy season (May and June), and atmospheric conditions in the mid- to late- rainy season

(July to October) and cool-dry season [46]. However, SWAT simulated ET for two land use maps (1994 and 2005) with the same climatic condition, soil data, and model parameters. Reductions of LAI due to forest conversion to plantations was the main cause of decreased ET.

There were large discrepancies in the SW, ET, and GWQ comparison results between the two stations, Ta Lai and Ta Pao. ET and SW were highly affected by seasonal variations at Ta Lai than Ta Pao station. On the other hand, GWQ remained unaffected in the rainy season and only decreased by approximately 5% in the dry season. To investigate the effects of LUCC in Dong Nai river subbasin (DN) (Ta Lai station) and La Nga river subbasin (LN) (Ta Pao station), we compared a fraction of main land use in 1994 and 2005. Subbasin areas of DN and LN were 10,639 km² (72.35% of the study area) and 4067 km² (27.65% of the study area), respectively. Figure 7 shows the fraction of land use in DN and LN. Forest (dense and sparse) and agriculture (orchard and crop) were the dominant land use patterns, occupying about 78.2% and 17.8%, respectively, of total DN area, and 59.7% and 38.5% of the total LN area, respectively, in 1994. The ratio of two land use types in 2005 changed in the range of 38.7–57.6% at DN and 36.0–61.0% at TL. In other words, forest area decreased by 26.4% and 39.7% from 1994 to 2005. Conversely, agricultural land increased by 117.5% and 58.3% at DN and LN, respectively. Despite similarities in land use change trends between the two subbasins, there is a difference in the catchment scale, the proportion of forest and agricultural land, and ratio of changed areas of each subbasin. The area of lost forest is less in DN than LN; contrastingly, the area of plantation-replaced-forest is much larger in DN than LN. This suggests that the amount of decrease in ET and SW is proportional to the ratio of lost forest area, and as the areas of plantation-replaced-forest increased, reductions in the infiltration rate lowered. In addition, the conversion from forest to agriculture in DN mainly occurred in the headwater area in Lam Dong and Dak Nong provinces. Meanwhile, this conversion also appeared in LN in the headwater area, Lam Dong province, and was scattered in Binh Thuan and Dong Nai provinces at the downstream (Figure 2). The spatial distribution of LUCC also affected the hydrology.

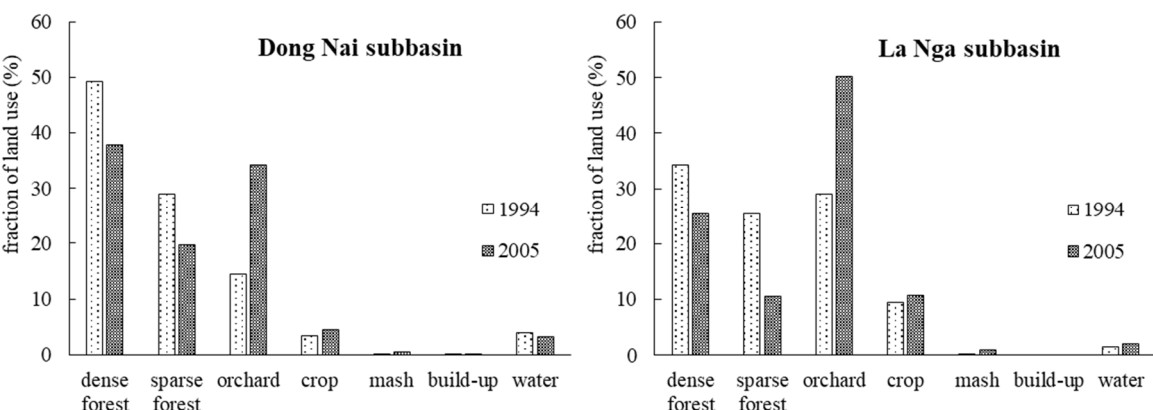

**Figure 7.** Fraction of land uses in Dong Nai and Ta Lai river subbasins in 1994 and 2005 (calculated from [2]).

## 4. Discussions

The results of this study validate the previously reported results, which state that land use changes involving conversion of forests to agricultural expansions increase the surface runoff due to reduced evapotranspiration and infiltration. While studying the dam functions in forests, Kuraji [47] collected data from experimental basins around the world and reported that forests have a positive effect on flood mitigation and a negative effect on drought mitigation. Deforestation and conversion to arable land/grassland are usually accompanied by an increase in surface runoff or total discharge [5,48–50]; particularly, forest replacement by annual crops in tropical rainforest regions tends to increase

annual streamflow quantities and storm flow events with higher peak discharges [51]. Surface runoff increases because of reduced evapotranspiration of the replaced vegetation due to the forest vegetation intercepts, which loses more water than other land use types [52,53]. Additionally, reductions in soil water content and groundwater discharge may be because the infiltration rate of forest land is higher than other land-use types [6].

Tan et al. [30] concluded that interception and infiltration rates were higher in forests than other land cover types; hence deforestation caused an increase in surface runoff and reduced water movement within the soil layers. Additionally, after deforestation or forest fires, the infiltration rates decreased due to the loss of tree cover. However, in the study area the lost forest was replaced immediately with crops and vegetation. Moreover, as discussed in [20], agricultural area with more shade trees were replaced quickly with secondary forests at any abandoned areas. Trees in these areas have higher transpiration rates outside the original forest. As a matter of fact, significant numbers of trees on agricultural land as well as secondary growth invading abandoned plots exist. This suggests that the soil was not thoroughly destroyed after deforestation and changes in infiltration were marginal. Infiltration reductions increase the streamflow during the rainy season; however, it is insufficient to reduce the dry season streamflow at Ta Lai station.

In the study area, irrigation of the coffee plantations directly affected the depletion of groundwater and fundamentally disrupted the regional hydrological system. Since coffee smallholders draw groundwater for dry season irrigation, the amount of water presently used exceeds the crop water requirement, as rightly stated, "smallholders irrigate more than twice the recommended level, with the belief that yield increases linearly with irrigation amount" [26,54,55]. As mentioned above, the results show SW and GWQ in the dry season reduced by 14.89% and 4.96%, respectively, at Ta Pao station (Figure 5) with the increase in the coffee plantation area. According to the previous studies [26,54–58], irrigation during the dry season is crucial to achieve high coffee yields as it assists in breaking flower bud dormancy and inducing fruit setting. Groundwater is the major source for coffee irrigation in the Central Highlands. The water supply for the coffee tree consists of a micro-basin irrigation system; every tree stands in a planting hole with dimensions of 2.6 m × 2.6 m × 0.2 m. The water is pumped up to the plantations and watered through a hose with 100 mm application depth. D'haezea et al. [53] concluded that the present groundwater abstraction at the Ea Tul watershed in the Dak Lak province is not sustainable in the dry year as it exceeds the safe aquifer yield. Controlled agricultural (coffee) expansion is necessary to solve the issue of water depletion and to ensure a sustainable environment.

## 5. Conclusions

Human activities, such as LUCC, especially conversion of forests to agricultural lands are the main factors that affect the streamflow regime. Quantifying the contribution of LUCC to the flow regime is important for water resources planning and management. In the present study area, forest coverage decreased from 73% in 1994 to 51% in 2005; in contrast, agricultural land increased from 24% in 1994 to 45% in 2005. The majority of land use conversions in the study area were due to expansion of coffee plantations. The impacts of LUCC on streamflow regime by assessing the changes of each hydrological component, such as actual evapotranspiration (ET), surface runoff (SUR), soil water content (SW), and ground water discharge (GW) were estimated based on the SWAT model. The significant findings from the analysis in this study were: (1) Deforestation and conversion to agriculture increases SUR drastically due to reductions in ET, (2) in contrast, SW and GWQ tend to decrease due to reduced infiltration after land cover change, (3) these reductions eventually increase in proportion; subsequently, decreasing the streamflow in the dry season when the proportion of agricultural land is higher than forest cover, (4) uncontrolled expansion of coffee plantations in conjunction with the existing coffee irrigation system will lead to severe water imbalances in the future.

**Author Contributions:** Conceptualization and methodology, N.C.Q.T. and A.K.; analysis, N.C.Q.T. and D.N.K.; data curation, H.Q.N.; writing—original draft preparation, N.C.Q.T.; writing—review and editing, D.N.K. and H.Q.N. All authors have read and agreed to the published version of the manuscript.

**Funding:** This research received no external funding.

**Institutional Review Board Statement:** Not applicable.

**Informed Consent Statement:** Not applicable.

**Data Availability Statement:** Not applicable.

**Acknowledgments:** We would like to thank the Center for Environmental Remote Sensing, Chiba University for supporting me as a cooperative researcher.

**Conflicts of Interest:** The authors declare no conflict of interest.

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
