# Peer review of "Impact of Forest Conversion to Agriculture on Hydrologic Regime in the Large Basin in Vietnam"

_water, doi:10.3390/w14060854_

Round 1
Reviewer 1 Report
This study has assessed the impact of conversion of forest into agriculture on the watershed hydrology of Dong Nai river basin in Vietnam. The study has used SWAT model for this purpose.
Following are my comments:
- First and foremost, I think the paper has been translated into English language. I would request the authors to correct it by some native English expert translator.
- The modus operandi of a standard english paper writing is missing in the paper.
- Introduction must start with a global perspective. Then local
- The review of literature in the introduction part is not weel organized. Authors are requested to organise the papers within the paragraph on the basis of a specific theme/ topic.
- Figure needs to be redrawn showing the World, Vietnam and from that, the study area (UDNB).
- How has been LULC generated, No methodology shown? Which Landsat Imagery, its resolution, classification method, accuracy assessment, nothing is discussed?
- What is the time step observational discharge data availble with the authorities (hourly, daily, monthly)?
- At what time step the model has been run?
- A graph showing average monthly discharge observational data at the place of Figure 2 is highly recommended for understanding the regime of the watershed.
- The authors are suggested to structure the results and discussion section into sub sections for better clarity of findings.
- I would recommend to separate the results and discussion sections.
Author Response
Response to Reviewer 1 Comments
- First and foremost, I think the paper has been translated into English language. I would request the authors to correct it by some native English expert translator. 2. The modus operandi of a standard english paper writing is missing in the paper.
Thank you very much for your comment. The manuscript has been edited by Taylor & Francis English Editing Services before submission. The revised version was also checked again by a native English-speaking colleague.
- Introduction must start with a global perspective. Then local
Thank you very much for your suggestion. In the "Introduction" part, first, we want to present the statement of the problem about land use change in Vietnam, and the scientific question is how this change affects the water cycle. Next is the literature review.
- The review of literature in the introduction part is not weel organized. Authors are requested to organise the papers within the paragraph on the basis of a specific theme/ topic.
Thank you very much for your suggestion. We rearranged the "literature review" part into separate themes based on the results of previous studies (lines 72-114).
- Figure needs to be redrawn showing the World, Vietnam and from that, the study area (UDNB).
Thank you very much for your suggestion. We edited the Figure 1.
- How has been LULC generated, No methodology shown? Which Landsat Imagery, its resolution, classification method, accuracy assessment, nothing is discussed?
Thank you very much for your questioning. Land use data used in this study are the results of Truong et al. 2018. We have added a description of Landsat imageries (lines 161-162).
Truong, N.C.Q.; Nguyen, H.Q.; Kondoh, A. Land Use and Land Cover Changes and Their Effect on the Flow Regime in the Upstream Dong Nai River Basin, Vietnam. Water, 2018, 10, 1206.
- What is the time step observational discharge data availble with the authorities (hourly, daily, monthly)?
Thank you very much for your questioning. Discharge data were collected originally in hourly time steps. It was processed by a local agency daily, monthly, and yearly according to a requested organization. We obtained daily data for this study.
- At what time step the model has been run?
The model has been run by daily time step. We have added at line 245.
- A graph showing average monthly discharge observational data at the place of Figure 2 is highly recommended for understanding the regime of the watershed.
Thank you very much for your suggestion. We have added the monthly discharge observational data (in comparison to precipitation data) in Figure 3.
- The authors are suggested to structure the results and discussion section into sub sections for better clarity of findings.
- I would recommend to separate the results and discussion sections.
Thank you very much for your suggestion. We have separated the results and discussion sections.

Reviewer 2 Report
Dear Editor.
I have finished my review on the proposed paper “Impact of Forest Conversion to Agriculture on Hydrologic Regime in the Large Basin in Vietnam” water-1609181-peer-review-v1.
Summary of the manuscript:
In the proposed paper, the authors’ goal is to evaluated the effect of land-use changes that took place during 1994 to 2005, on runoff, actual evapotranspiration, soil water content and ground water in Vietnam. They used SWAT model to quantify these changes, using observed data of rainfall and river discharge. The results showed that the land use changes greatly impacted the forest coverage of the study area and the surface runoff.
General review:
- Generally, the manuscript presents a very interesting topic and the specific research seems to include some significant points for the research community of this field.
- The proposed paper is very well written with very good use of English language. Except some minor grammatical mistakes and word errors, this paper is written with a very good scientific style. The authors should check again the paper to correct these minor mistakes.
- The proposed paper is very well structured. It begins with an analytical Introduction with the appropriate references that helps the reader to get into the subject immediately. In Introduction there is an effort to provide previous studies with similar scientific content, which took place in the research area and in some cases in other countries. Authors describe and set very well the scientific problem and how other researchers have approached. At the end of Introduction, authors clearly state the goals of the research.
- The methodology is generally very interesting, and well explained, so other researchers could easily repeat it. Every aspect of methodology is well documented with the use of the appropriate literature. However, I have some concerns, which are explained below.
- The results scientifically explained with the use of the appropriate scientific literature.
- The quality of the work in Results and Discussion section is very qualitative.
- Conclusions are appropriate for this paper.
- The authors did not provide any new methodology. Although the methodology is very interesting, however is not novel, as it incorporates already known and existing models and procedures.
Points for revision:
In my opinion, the proposed paper could be characterized as a very good research work, complies with aims of WATER.
Nevertheless, I have some points for revision.
Lines 66-67: Here, you highlight a very significant aspect that influence the runoff and flood generation in forest environments. “It is known that forests present finite capabilities to retain large amounts of precipitation, especially during extreme rainfall events, even if the forest cover percentage is significantly high” (Kastridis et al. 2021, De Jong 2016). So, the runoff and flood are highly influenced from rainfall intensity and the forest cover sometimes is not capable to retain the rainfall, independently if the forests are located in developing or developed countries. Please, rephrase these two lines in the text addressing the above, and add the proposed 2 literatures with the existing.
Lines 238-242 and table 2: Here, you presented the known Curve number II (CN2). In lines 129-134 you informed us that the study area has two seasons (dry and rainy). However, the presented calculation of CNII is for standard initial soil moisture conditions. More specifically, you calculated the CNII for initial loss rate of 20%, which is the reference value and corresponds to average humidity conditions (Antecedent Moisture Conditions, (AMC II)). But, throughout the year the AMC is changing in your study area (dry and rainy season). The runoff generation is significantly different during the rainy period when the soil moisture is higher, in comparison to the drought season when the soil moisture in lower. For that reason, when CN is estimated the AMC is taken into account according the following table (Chow et al. 1988):
Classification of antecedent moisture condition classes (AMC) for the SCS method of rainfall abstractions (source: Chow et al. 1988; table 5.5.1, p. 149).
AMC group |
Total 5-day antecedent rainfall (mm) |
|
Dormant season |
Growing season |
|
I |
Less than 13 |
Less than 35 |
II |
13 to 28 |
35 to 53 |
III |
Over 28 |
Over 53 |
The value that corresponds to the average humidity conditions (AMC II) is related to the other two typical initial soil moisture conditions (AMC I and AMC III), according to the following empirical relationships (Chow et al. 1988):
CNI = 4.2 CNII/10 − 0.058 CNII (2)
CNIII = 23 CNII/10 + 0.13 CNII
You calculated the runoff with the same value of CN (CNII) for all the seasons of the year. I think that you should make some changes to your results.
Table 3. There is a small mistake in row 4, NSE column, “0.50 < NS. E ≤ 0.65”.
Lines 253-282 and tables 3 and 4: I think that you could provide also the RMSE (that you have already calculated) with the Standard Deviation (SD), in order to give an in-depth view of model performance. The best way to interpret the RMSE and understand if is acceptable for model evaluation, is to provide the Standard Deviation (SD). It is known from previous studies that RMSE values less than half of the SD (Standard Deviation) of the measured data may be considered low and acceptable for model evaluation (Kastridis et al. 2021, Singh et al. 2005). Provide the SD with the RMSE, discuss in the text the above mentioned, and explain if the RMSE is acceptable in terms of the SD, adding the proposed literature to support your discussion.
References
Chow V.T., Maidment D.R. and Mays L.W. (1988), Applied Hydrology, McGraw‐Hill: New York, NY, USA, 1988, p. 572, ISBN 0 07‐010810‐2.
De Jong, C. European perspectives on forest hydrology. In Forest Hydrology: Processes, Management and Assessment; Amatya, D., Williams, T., Bren, L., De Jong, C., Eds.; CABI:Wallingford, UK, 2016; pp. 69–87.
Kastridis, A.; Theodosiou, G.; Fotiadis, G. Investigation of Flood Management and Mitigation Measures in Ungauged NATURA Protected Watersheds. Hydrology 2021, 8, 170. https://doi.org/10.3390/hydrology8040170.
Singh, J., et al (2005). Hydrological modeling of the Iroquois river watershed using hspf and swat1. Journal of the American Water Resources Association, 41, 343–360. https://doi.org/10.1111/j.1752-1688.2005.tb03740.x.
Author Response
Response to Reviewer 2 Comments
- Lines 66-67: Here, you highlight a very significant aspect that influence the runoff and flood generation in forest environments. “It is known that forests present finite capabilities to retain large amounts of precipitation, especially during extreme rainfall events, even if the forest cover percentage is significantly high” (Kastridis et al. 2021, De Jong 2016). So, the runoff and flood are highly influenced from rainfall intensity and the forest cover sometimes is not capable to retain the rainfall, independently if the forests are located in developing or developed countries. Please, rephrase these two lines in the text addressing the above, and add the proposed 2 literatures with the existing.
Thank you very much for your suggestion. We have rephrased this content in line 63-71.
- Lines 238-242 and table 2: Here, you presented the known Curve number II (CN2). In lines 129-134 you informed us that the study area has two seasons (dry and rainy). However, the presented calculation of CNII is for standard initial soil moisture conditions. More specifically, you calculated the CNII for initial loss rate of 20%, which is the reference value and corresponds to average humidity conditions (Antecedent Moisture Conditions, (AMC II)). But, throughout the year the AMC is changing in your study area (dry and rainy season). The runoff generation is significantly different during the rainy period when the soil moisture is higher, in comparison to the drought season when the soil moisture in lower. For that reason, when CN is estimated the AMC is taken into account according the following table (Chow et al. 1988)
Thank you very much for your suggestion. We added the explanation of the issue of CN selection for different seasonality in line 257-263.
- Table 3. There is a small mistake in row 4, NSE column, “0.50 < NS. E ≤ 0.65”.
Thank you very much. We edited Table 3.
- Lines 253-282 and tables 3 and 4: I think that you could provide also the RMSE (that you have already calculated) with the Standard Deviation (SD), in order to give an in-depth view of model performance. The best way to interpret the RMSE and understand if is acceptable for model evaluation, is to provide the Standard Deviation (SD). It is known from previous studies that RMSE values less than half of the SD (Standard Deviation) of the measured data may be considered low and acceptable for model evaluation (Kastridis et al. 2021, Singh et al. 2005). Provide the SD with the RMSE, discuss in the text the above mentioned, and explain if the RMSE is acceptable in terms of the SD, adding the proposed literature to support your discussion.
Thank you very much for your suggestion. RSR is the ratio of RMSE and SD. According to that, less than 0.5 RSR value as the most stringent “very good” rating, and two less stringent ratings of 10% and 20% greater than this value for the “good” and “satisfactory” ratings, respectively (Table 3). We have added this description in 278-282.

Reviewer 3 Report
Dear Authors,
As per my view, authors did a good job to frame this article of global importance. Paper reveal the Impact of Forest Conversion to Agriculture on Hydrologic Regime in the Large Basin in Vietnam and falling under the scope of the journal and written well. Therefore, paper could be acceptable for publication in the present form. But if authors can also discuss a bit another others crop instead of only coffee plantations, it will be great interest.
Author Response
Response to Reviewer 3 Comments
As per my view, authors did a good job to frame this article of global importance. Paper reveal the Impact of Forest Conversion to Agriculture on Hydrologic Regime in the Large Basin in Vietnam and falling under the scope of the journal and written well. Therefore, paper could be acceptable for publication in the present form. But if authors can also discuss a bit another others crop instead of only coffee plantations, it will be great interest.
Thank you very much for your comments. We will try to evaluate the conversion of the other crops in future works.

Round 2
Reviewer 1 Report
Thank you for incorporating the suggestions.
Author Response
Dear Reviewer 1,
Thank you very much for your valuable comments and suggestions.
Best regards,
Reviewer 2 Report
Dear authors.
I have studied your responses to my comments. I am overall satisfied about your work in review process.
However, I do not agree about the explanation that you give, concerning the CNII. Because, I am not very familiar with the tropical climate, I will accept the statement in lines 382-388, which you supported with Dile et al. study.
I agree with you about the RSR and RMSE. You have explained the issue very well. Finally, there is a minor problem with the references [14] and [43], which are the same. In the place of reference [14] you should add Sapountzis et al. 2021 (doi.org/10.30955/gnj.003905).
Good luck.
Author Response
Dear Reviewer 2,
Thank you again for your comments, these were very helpful to improve the quality of the manuscript.
Response to Reviewer 2 Comments
There is a minor problem with the references [14] and [43], which are the same. In the place of reference [14] you should add Sapountzis et al. 2021 (doi.org/10.30955/gnj.003905).
We removed the reference [43]. We have studied Sapountzis et al. 2021. Thank you for your suggestion. It has been added in the place of reference [14].
Best regards,